# Macrophage-Targeted Punicalagin Nanoengineering to Alleviate Methotrexate-Induced Neutropenia: A Molecular Docking, DFT, and MD Simulation Analysis

**DOI:** 10.3390/molecules27186034

**Published:** 2022-09-16

**Authors:** Ritu Karwasra, Shaban Ahmad, Nagmi Bano, Sahar Qazi, Khalid Raza, Surender Singh, Saurabh Varma

**Affiliations:** 1ICMR-National Institute of Pathology, Safdarjung Hospital Campus, New Delhi 110029, India; 2Department of Computer Science, Jamia Millia Islamia, New Delhi 110025, India; 3Department of Pharmacology, All India Institute of Medical Science (AIIMS), New Delhi 110029, India

**Keywords:** punicalagin, mannose, nanoparticles, molecular docking, MD simulation

## Abstract

Punicalagin is the most bioactive pomegranate polyphenol with high antioxidant and free-radical scavenging activity and can potentially cure different ailments related to the cardiovascular system. The current research work was envisioned to predict the targeting efficiency of punicalagin (PG) nanoparticles to the macrophages, more specifically to bone marrow macrophages. For this, we selected mannose-decorated PLGA-punicalagin nanoparticles (Mn-PLGA-PG), and before formulating this nanocarrier in laboratory settings, we predicted the targeting efficiency of this nanocarrier by in silico analysis. The analysis proceeded with macrophage mannose receptors to be acquainted with the binding affinity and punicalagin-based nanocarrier interactions with this receptor. In silico docking studies of macrophage mannose receptors and punicalagin showed binding interactions on its surface. PG interacted with hydrogen bonds to the charged residue ASP668 and GLY666 and polar residue GLN760 of the Mn receptor. Mannose with a docking score of −5.811 Kcal/mol interacted with four hydrogen bonds and the mannose receptor of macrophage, and in PLGA, it showed a −4.334 Kcal/mol docking score. Further, the analysis proceeded with density functional theory analysis (DFT) and HOMO–LUMO analysis, followed by an extensive 100 ns molecular dynamics simulation to analyse the trajectories showing the slightest deviation and fluctuation. While analysing the ligand and protein interaction, a wonderful interaction was found among the atoms of the ligand and protein residues. This computational study confirms that this nanocarrier could be a promising lead molecule to regulate the incidence of drug-induced neutropenia. Furthermore, experimental validation is required before this can be stated with complete confidence or before human use.

## 1. Introduction

The nanoparticulate system has gained widespread use in chemotherapeutics and is currently involved in the neurodegenerative, cardiovascular system, cancer, and other chronic ailments. The nanoparticulate drug delivery system represents a viable option that provides the spatial and temporal controlled delivery of peptides, therapeutics, or other bio-actives [1]. Numerous researchers have proved that the nanoparticles have improved cancer treatment through sophisticated functionalisation and their ability to accumulate within certain tumours [2]. Despite all this, several nanomedicines fail during clinical trials due to specific issues. One of them is a lack of understanding of nanoparticle design, or how the design of the nanoparticles impacts their ability to cross transport barriers such as circulation in the bloodstream, extravasation in tumour cells or transport into the tumour tissues, internalisation in tumour tissue, or the release of active cargo in situ. Computational tools and multi-scale simulations allow us to design the nanoparticles or study their effect on biological transporters, drug release, or targeting specific receptors in biological scenarios [3]. Currently, computational modelling and multi-scale simulations can predict a range of parameters influencing nanoparticles in the biological scenario. This type of future pipeline enables us to provide general design principles that, when combined with patient-specific data, provide the information related to personalised treatment and care. In addition, in silico analysis can minimise the expenses involved in the laboratory’s more conventional trial and error procedures, especially when pooled with the latest machine learning techniques such as active learning [4].

Copious reports available in scientific databases have reported that many computational methods and models can predict diverse phases of tumour initiation and growth and the interaction of nanoparticles within the body and the tumour [5]. During tumour treatment, certain medications in chemotherapy are provided to the patient to improve their quality of life. However, at a certain point, these medications offer some side effects that impact the patient’s health status. One such anticancer drug, methotrexate, causes several side effects, but the most prominent side effect encountered is neutropenia, i.e., lowering the level of neutrophils in blood circulation [6]. Methotrexate is more commonly prescribed for treating Rheumatoid arthritis as a disease-modifying antirheumatic drug and is often accompanied by haematological toxicities (thrombocytopenia, leukopenia, pancytopenia, megaloblastic anaemia), even at small doses [7]. This is a severe concern pertaining to chemotherapeutic patients as well as to autoimmune RA patients. Therefore, stepping into this, filgrastim is frequently co-prescribed with chemotherapeutic drugs at risk of developing neutropenic conditions [8]. Looking more into the solution segment, we came across punicalagin, a phytoconstituent of pomegranate which is also effective in healing neutropenic conditions [9]. Punicalagin (PG) is a phenolic compound that has been reported to exert numerous beneficial effects in the treatment of certain disorders such as cancer [10], inflammatory disorders [11], nephroprotective [12], and cardiovascular diseases, to scavenge oxidative free radicals, and to reduce the risk of atherosclerosis [13] and other specific chronic ailments.

In the current research work, we selected the mannose-decorated poly (lactic-co-glycolic acid) or PLGA and punicalagin (Mn-PLGA-PG) nanoparticles to alleviate methotrexate-induced neutropenic settings. The selection of this nanocarrier was based on the ideology that neutropenia occurs in the bone marrow; therefore, to counteract this condition, we have to target our punicalagin molecule extensively to the bone marrow site. Bone marrow macrophages have abundant mannose receptors on their surface, and if we decorate the PG nanocarriers with mannose, they bind extensively to bone marrow macrophages. From an extensive literature search and to the best of the authors’ information, none of the existing studies have described the targeting efficiency of Mn-PLGA-PG nanocarriers to bone marrow, which is responsible for alleviating methotrexate-induced neutropenia. Therefore, in this study, we explored the design of a targeted nanocarrier for bone marrow and whether the proposed nanocarrier efficiently delivers the PG molecule to its site of action by molecular docking and dynamics simulation studies. Further, the study is in an experimental condition for synthesis and characterisation to design and validate the computational results.

## 2. Methodology

The methods we followed for the complete study comprised multiple steps, which are shown in Figure 1. Further, the detailed methods are as follows.

### 2.1. Protein and Ligand Preparation

Researchers have already elucidated the presence of mannose receptor sites on macrophages; therefore, after the validation of previous studies, we took PDB ID 1EGI from https://www.rcsb.org/ (accessed on 10 August 2022) for the macrophage mannose receptor [14]. Along with chain A, solvents and other metals/ions were removed, and the longest chain B was then taken further for the studies as both chains are dimer [14]. After the fixation of the problem with Cα (802), we removed the water beyond 3.0 Å. As far as ligands are concerned, we took punicalagin (CID44584733), mannose (CID18950), poly (lactic-co-glycolic acid), or PLGA (CID23111554) and prepared them for the molecular docking in the 3D SDF format.

### 2.2. Molecular Docking Studies

A molecular docking study was conducted to investigate if the ligand (PG) bound to the macrophage receptor and the behaviour of other carriers’ mannose and PLGA. We performed molecular docking studies by AutoDock (http://autodock.scripps.edu/) (accessed on 10 August 2022) and MGL tools (http://mgltools.scripps.edu/) (accessed on 10 August 2022) [15]. The grid file was generated around the complete chain B for blind docking studies as it was not bound to any native ligand. All three compounds were docked individually as they must be delivered at the target site and unbind to work against their intended purpose. Further, the academic version of Schrodinger Maestro (https://www.deshawresearch.com/downloads/download_desmond.cgi/) (accessed on 10 August 2022) [16] and PyMol (https://pymol.org/2/) (accessed on 10 August 2022) [17] was used for the visualisation of interaction and the bonding analysis of the protein–ligand complexes.

### 2.3. QM-Based Density Functional Theory Analysis

The quantum mechanics-based density functional theory (DFT) analysis, highest occupied molecular orbital (HOMO), and lowest unoccupied molecular orbital (LUMO) analysis were performed with the jaguar optimisation tool in the Schrodinger maestro [18]. The B3LYP-D3 theory with the basic set of 6-31G** was used with a Self-consistent field (SCF) spin in automatics, and grid density in the medium was selected for the density functional theory [19]. In the SCF tab, a quick accuracy level with an atomic overlap in initial guess along with the 48-maximum iteration, 5 × 10^−5^ Hartree of energy change, and 5 × 10^−6^ of RMS density matrix change were kept. Optimising 100 steps with analytic integrals near convergence with the Schlegel guess at initial hessian and redundant internal coordinates was kept. The PBF solvent model with water solvent in the optimised gas-phase structure was chosen for the vibration frequencies and surfaces of molecular orbitals, density, and potential were taken where the following calculations, namely electrostatic potential, average local ionisation energy (kcal/mol), non-covalent interaction (with 20 pts/Å of grid density), electron density, spin density, molecular orbitals for Alpha (from HOMO^−^ 0 to LUMO^+^ 0), and the Beta (from HOMO^−^ 0 to LUMO^+^ 0) with a total number of 2 orbitals in both cases were kept.

### 2.4. Molecular Dynamics Simulations

The dynamic behaviour of the protein–ligand complex in simulated physiological conditions was studied with a molecular dynamic (MD) simulation. MD simulations of the macrophage mannose receptor–ligand complex were performed using the academic version of the Desmond application available with Schrodinger maestro (v 2020-4) [16,17]. The macrophage mannose receptor–punicalagin (2337 atoms) was solvated in a distance of 10 × 10 × 10 Å orthorhombic periodic box built with the SPC water model. The whole system was neutralised by adding 4Na^+^ counter ions, and the solvated system was energy minimised and position restrained with the OPLS4 force field [20,21]. For macrophage mannose receptor–mannose (2258 atoms), 1Na^+^ and macrophage mannose receptor–PLGA (2251), 2Na^+^ was added to neutralise the system. Both systems were also solvated in a 10 × 10 × 10 Å orthorhombic periodic box built with the SPC water model and minimised with the same OPLS4 forcefield [22]. Ions and salt placement within 20 Å were excluded in all systems. Further, 100 ns of a molecular dynamics simulation was carried out at 1 atm pressure and temperature of 300 K, including the NPT ensemble with a recording interval of 100 ps, resulting in 1000 reading frames for each complex separately to check and evaluate the behaviour [23]. In the end, various parameters of the molecular dynamics simulation study, such as fluctuation, deviation, protein–ligand contacts, and ligand binding site were analysed to check the structural deviation fluctuations, compactness, stability, and protein–ligand interactions within the solvated system during the simulation time [24,25].

## 3. Results

### 3.1. Interaction Analysis

The molecular docking studies of mannose receptors with punicalagin, PLGA, and mannose showed potential confidence in generating a nano-complex for all three ligands, and it could be provided as a dose for targeted therapy after complete validation. In Figure 2A, we have shown the macrophage mannose receptor with the docked pose in the complex with mannose with a docking score of −5.811 Kcal/mol and interacting with four hydrogen bonds to the mannose receptor of the macrophage with two negatively charged residues GLU706 and GLU719 and also with two hydrophobic residues VAL716 and PHE708. In silico docking studies of the macrophage mannose receptor and punicalagin showed a binding interaction on its surface with a docking score of −4.00 Kcal/mol. In Figure 2B, the PLGA with the macrophage mannose receptor showed a −4.334 Kcal/mol docking score and it interacted with two hydrogen bonds, LEU694 (hydrophobic) and HIS692 (polar), and one salt bridge was formed among the O^−^ atoms of the PLGA and LYS652 of the protein. In Figure 2C, we have demonstrated the interaction of punicalagin with the macrophage mannose receptors. Punicalagin formed six hydrogen bonds with different amino acids of the macrophage mannose receptor. The polar residue GLN760 formed two hydrogen bonds with O^−^ and O atoms while the positively charged amino acid ARG663 interacted with the O atom and OH atom, and the OH atoms formed another hydrogen bond with negatively charged amino acid ASP668. Another OH formed one hydrogen bond with GLY666. Further description in Table 1 is provided to better understand the energies and bonds with the molecular docking score and the generated energy with the respective ligands.

### 3.2. Density Functional Theory and HOMO–LUMO Analysis

The density functional theory is a computational quantum mechanics-based model used to investigate the electronic structure of multi-atomic systems, such as ligand molecules. The DFT is a core physics concept that computational chemists have taken, and now it is also used in drug designing to investigate the stability of the drug candidates. In this analysis, we performed multi-iterative calculations to discover the relative and other energies and the HOMO and LUMO sites of the ligand molecules. The QM-convergence monitor was used for the graphical representation and calculative understanding of the relative energy. In Figure 3A, we have shown the QM-convergence of the mannose that has shown its 0 relative energy in a short time that started almost at 0.05, and in just 15 iterations, it went down to 0, meaning the compound is very much bindable with assuring stability. This is because the compound size is petite, and this shows that mannose can be one of the best natural carriers for the compound and will not impose any significant side effects. Moreover, for the PLGA, the relative energy started at 0.14, went down to 0 in just 30 iterations, and performed smoothly afterwards. Moreover, in this case, the sudden fall in relative energy was noticed at the 28th iteration (Figure 3B). In Figure 3C, we have shown the convergence of the punicalagin compound, our drug compound. In this case, the slow fall of the relative energy was noticed, but at the 48th iteration, it suddenly fell to zero after a slight deviation. Despite this, after 50 iterations, it showed a very smooth performance.

In Figure 3, other than the relative energy, we have also shown other convergences that can be used for other calculative measures. Further, Figure 4 shows the HOMO and LUMO sites of the compounds that best describe how efficiently the compounds will be bonded in the synthesis process and how efficiently these bonds will be broken in the delivery step at macrophages. Further, on this behalf, the entire decision for the synthesis has been taken.

### 3.3. Molecular Dynamics Simulation

MD simulation provides information about the receptor–ligand complex with the timescale, so we performed the MD simulation for 100 ns on all three complexes generated through molecular docking. Molecular dynamic simulations of the protein–ligand complex were performed using the Desmond package, and we analysed the trajectory files for root mean square deviation and root means square fluctuation simulative interactions. The details are as follows.

#### 3.3.1. Root Mean Square Deviation

Root mean square deviation (RMSD) describes the average change in the displacement of an atom in a specific molecular level confirmation with its referential confirmation data. The trajectory analysis for the complex of the macrophage mannose receptor and mannose both showed a higher deviation because of the mannose structure. Mannose is a highly flexible sugar that usually shows higher fluctuations. The initial fluctuations were noticed due to the initial head and change in the medium. The protein and ligand showed a deviation of 2.63 Å and 6.35 Å at 0.90 ns, respectively, and it went off to 4.56 Å and 76.77 Å at 13 nanoseconds. After these fluctuations, the protein and ligand showed some stability, meaning fewer fluctuations than the initial fluctuations, and till 100 ns, it went down to 3.44 Å and 66.25 Å for the protein and ligand, respectively, which was lower compared to the deviation at 13 ns. The protein deviation in this condition was average after the initial heat-based deviation is ignored, but the ligand showed much deviation as most of its end contained OH atoms that were very rotatable (Figure 5A). The macrophage mannose receptor in a complex with the PLGA also showed a considerable deviation as it is a polymer and its expected behaviour would be to show the deviation at most of the conditions as the bonds are interconnected and rotatable. The initial deviation in 0.90 ns for the protein was 1.76 Å, and for the PLGA, it was 3.38 Å, and it continued to deviate till 61 ns and after it showed some stability but at 96 ns, it fell to 7.57 Å for the ligand. In contrast, the protein deviation was quite good and showed a stable performance after the initial deviation; if the initial deviation can be ignored, the overall deviation will become almost zero meaning the protein is stable. However, due to the poly nature of the ligand, it showed deviation at some extinct (Figure 5B). Macrophage mannose receptors in complex with punicalagin did not show much deviation, meaning the complex was stable and had good binding potential to the target. The initial deviation was because of the heat and heavy size of the compound, but after the initial deviation, the complex showed a stable performance. In the initial 8 ns, the protein showed a deviation of 8.35 Å while the ligand showed a deviation of 4.19 Å and at 16 ns it went to the highest and showed a deviation in the protein case of 9.05 Å, and for the protein case, it showed 5.43 Å. However, after 16 ns till 100 ns, the deviation was ignorable; for the protein, it showed a deviation of 8.17 Å, and for the ligand, it showed 4.33 Å. If the initial deviation is ignored, both the protein and the ligand showed a deviation less than 2 Å that is acceptable for the protein–ligand complex (Figure 5C).

#### 3.3.2. Root Mean Square Fluctuations

The root mean square fluctuations are the measures to know about the residual fluctuations, and they also provided information about how each residue at which nanoseconds fluctuated from its original position. ILE625 to LYS627, GLU706 to GLU733 and PRO7676 showed fluctuations of more than 2 Å but also a colossal contact among the protein residues and atoms of the ligand. ILE625, LYS627, CYS628, PRO629, GLU630, ASP631, TRP632, ALA634, SER635, THR638, LYS643, ALA646, LYS647, LYS649, HIS650, GLU651, LYS652, LYS653 THR654, PHE656, GLU657, ASP660, ARG663, ALA664, LEU665, GLY666, GLN679, TRP682, ARG683, THR686, ALA687, SER688, GLY689, SER690, TYR691, HIS692, LYS692, LYS693, SER703, PRO704, SER705, GLU706, GLY707, PHE708, SER711, ASP712, GLY713, SER714, VAL716, SER717, TYR718, GLU719, ASN720, TRP721, ALA722, TYR723, GLU725, ASN727, ASN728, GLU733, TYR734, ASN750, GLU752, HIS753, ASN756, GLN762, LYS763, GLN765, THR766, and PRO767 were the residues that interacted with the mannose and the huge interaction also confirmed the suitability of the mannose receptor at the macrophage (Figure 6A). The fluctuation of the macrophage mannose receptor and PLGA complex was not much, but the ligand contact was quite good. SER238 to GLN730 showed fluctuations beyond 2 Å, and the interacting residues were ILE625, LYS627, CYS628, PRO629, GLU630, ARG637, SER639, ALA646, LYS647, GLY648, LYS649, HIS650, GLU651, LYS652, ARG663, SER671, ILE672, ASN673, ASN674, LYS675, GLU676, GLN678, GLN679, TRP682, ARG683, THR686, SER688, GLY689, SER690, TYR691, HIS692, LYS693, LEU694, LEU699, SER703, SER705, GLU719, ASN720, TYR729, ASN731, LEU738, LYS739, ASP741, PRO742, THR743, MET744, TRP746, LEU754, ASN755, ASN756, GLN762, GLY764, GLN765, THR766, and PRO767 (Figure 6B).

The macrophage mannose receptor with the punicalagin showed fewer fluctuations but a good number of contacts. TYR701 to ASN731 and GLY764 to PRO767 showed fluctuations beyond 2 Å while the interacting residues were SER639, LEU640, ARG659, ARG663, GLY666, GLY667, ASP668, LEU669, SER671, ASN673, ASN674, GLU677, GLY698, THR700, TYR701, GLY702, SER703, PRO704, SER705, GLU706, GLY707, PHE708, THR709, TRP710, SER711, ASP712, GLY713, SER714, TYR718, ASN720, TRP721, GLN760, GLN762, LYS763, GLY764, GLN756, THR766, and PRO767 (Figure 6C). The overall interaction count for all three conditions was huge and showed a tremendous interaction distribution, meaning the compound will be bound and represents a good interaction during the in vitro and in vivo (in vitvo) validation.

#### 3.3.3. Simulative Interaction Analysis

The simulative interaction analysis showed the residues interacting with the ligand during the simulation time with time durations. Two hydrophobic residues, VAL716 and PHE708, interacted with the OH atoms, while the polar residue SER705 had one interaction with the OH atom. The positively charged amino acid LYS653 interacted with two individual OH atoms, while the negatively charged amino acids GLU706, GLU656, GLY725, and GLY752 interacted with two OH atoms of the mannose (Figure 7A). The PLGA in complex with the macrophage mannose receptor showed a good web of interaction with its atoms. The positively charged amino acids LYS693, LYS652, ARG637, LYS675, and LYS627 formed various interactions with OH atoms, and a few water bridges were also formed. The hydrophobic residues MET744, LEU694, PRO742, and ILE672, interacted with the O and OH atoms, while the polar residues LYS627, LYS675, ARG637, LYS693, and LYS652 interacted with the OH, O, and O^−^ atoms of the PLGA. There were 14 water molecules involved in forming the water bridges among the protein residues and the PLGA (Figure 7B). The punicalagin showed a good interaction during the simulative period by forming the water bridges, pi-cation, and hydrogen bonds. The positively charged amino acids ARG663 and ARG659 formed various direct interactions, a few water bridges, and one pi-cation with the benzene ring, while the negatively charged amino acids ASP712, ASP668, and GLU677 also formed water bridges and direct interactions. The polar residues GLN765, THR766, SER711, SER639, ASN673, THR709, GLN762, GLN760, and SER793 formed various water bridges and hydrogen bonds. Three hydrophobic residues, TRP710, LEU640, and LEU669, were also involved in the water bridges, while the GLY707, GLY702, GLY698, and GLY667 formed the water bridges and other interactions with the various atoms of the punicalagin (Figure 7C). Further, the count of interactions is provided in Figure 8 for better understanding.

## 4. Discussion

In silico analyses have highlighted the best possible design for bone marrow-targeted nanoparticles. In our scientific investigation, we used punicalagin, mannose, and PLGA to perform the molecular docking and to ensure the complex behaviour, we executed the molecular 100 ns dynamics simulation on the same generated complexes. The targeted nanoparticle delivery system offers certain advantages in terms of the site-specific delivery of cargo. This present work is based on the idea of an active targeting approach in which a ligand molecule is decorated on nanoparticles encapsulated with drug moiety [1]. These targeted nanocarriers mediate the receptor-oriented endocytosis process, thereby delivering the drug (PG) to bone marrow (site of action). These targeted delivery systems enable the cargo to release at the specific site and avoid their peripheral deposition. When formulating a nanocarrier, overcoming nanoparticle transport barriers, tissue penetration, or cargo release at a specific site is of utmost importance. Computational in silico analysis allowed a fast and systematic exploration of nanoparticles designed to deliver the therapeutic drug at the site of action [3,4]. General principles and guidelines from such tools and techniques have helped design effective treatments [26]. The study by Gupta and Rai, 2018 reported that the permeation of nanoparticles through skin layers is predicted with the help of constrained and unconstrained molecular dynamics simulations [27]. This study corroborates with our research work in which the targeting efficiency was investigated with target receptors. Other research encountered used machine learning techniques for predicting parameters of nanoparticles.

Yan 2019 et al. conducted in silico profiling of nanoparticles using universal nano descriptors [27]. Lunnoo et al. 2019 reported an in silico study of gold nanoparticle uptake in mammalian cells and predicted nanoparticle parameters such as size, shape, surface charge, and aggregation [28]. Specific elucidation can be provided to tailor nanoparticles and design them according to patient needs, such as personalised medicine. Constructing useful in silico tools will require close validation with in vitro and in vivo results. In this pandemic era, Weiss et al., 2020 reviewed the nanotechnology-based approaches against COVID-19 [29,30,31]. Qu et al. 2020 described the atomistic simulations between nanoparticles and lipid bilayers [32]. Stillman et al. 2020 described numerous methods of in silico modelling for cancer nanomedicines [32]. Comparing the aspects of molecular designing and validation with multiple computational algorithms that we performed in this study has now been in trend for its synthesis in the chemical lab and in vitro or in vivo validation.

Similarly, Casalin et al., 2019 described molecular modelling for nanomaterials; their challenges and perspectives were also discussed in detail [33]. The overarching aim was to design bone marrow macrophage-targeted nanoparticles of punicalagin for treating chemotherapeutic drug-induced neutropenia. Effective nanoparticle design requires an organised method for prototyping. In silico modelling has advanced to be an effective tool that minimises expensive trial-error procedures. Although such modelling does not exist in isolation, a collaboration between experimentalists, mathematical modellers, and clinicians will be required to inform the preeminent knowledge transfer. Initially, as in the first step, this methodology will help identify guidelines for designing suitable nanoparticles. We envisage a pipeline where the theoretical predictions are verified against the clinical effects and then returned to inform about future simulations [26].

Moving further to our research, the molecular docking and MD simulation studies predicted the targeting efficiency of mannose-decorated PLGA-punicalagin nanoparticles (Mn-PLGA-PG). The molecular docking analysis gave us a good docking score with all three compounds, Punicalagin (drug candidate) and the other two biodegradable nanocarriers bound to the surface of macrophages mannose receptor. MD simulations were carried out for 100 ns to analyse the compactness, fluctuations, residue interactions and, most importantly, protein–ligand stability. In the presence of the ligand, the stability of the proteins increased due to proper bonding coordination. Protein RMSD was almost the same for all ligands, and although the ligands RMSD varied, all were below 1.5 Å. However, the ligands’ fit on the protein entirely deviated. The RMSD of the ligand fit on the protein started stabilising with the timescale. We also noticed the presence of significant hydrogen bonds, water bridges, and hydrophobic interactions during the simulation period. Moreover, there were enough interactions shown by all complexes during the simulation timescale. The findings depicted that the identified complex (Mn–PG) could be a potential lead candidate to target the mannose receptor on the surface of bone marrow macrophages by interacting at the binding pocket. These simulation studies thereby enabled us to predict the design of nanoparticles in biological settings [27]. The method of employing in silico tools to predict the targeting efficiency before any experimental study is of utmost importance to the pharmaceutical industry. Molecular modelling techniques such as molecular dynamics and simulations can predict the drug affinity and binding energy for the polymer matrix PLGA. This shows that drug loading and targeting in the nanocarriers can be estimated, saving valuable time and resources for the experimentalists.

## 5. Conclusions

The findings suggest that the engineering of NPs with the aid of computational tools and techniques helps to reduce the time and cost associated with their design, development, and deployment. The molecular docking, DFT analysis, and MD simulation analysis provided us with a lot of information and clues to proceed with experimental designs. This could lead to the development of an efficient carrier system for punicalagin with a macrophage-targeting approach. Our futuristic work will synthesise this nanocarrier in laboratory settings and evaluate its effect in the biological scenario.

## Figures and Tables

**Figure 1 molecules-27-06034-f001:**
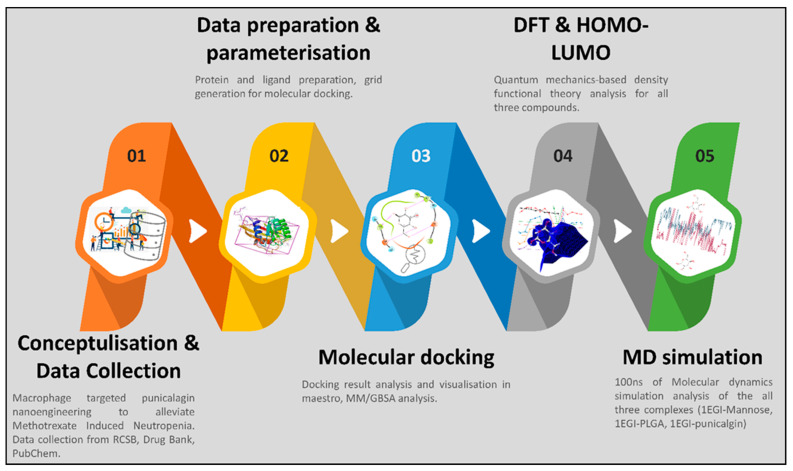
The graphical abstract of the methods for the complete study.

**Figure 2 molecules-27-06034-f002:**
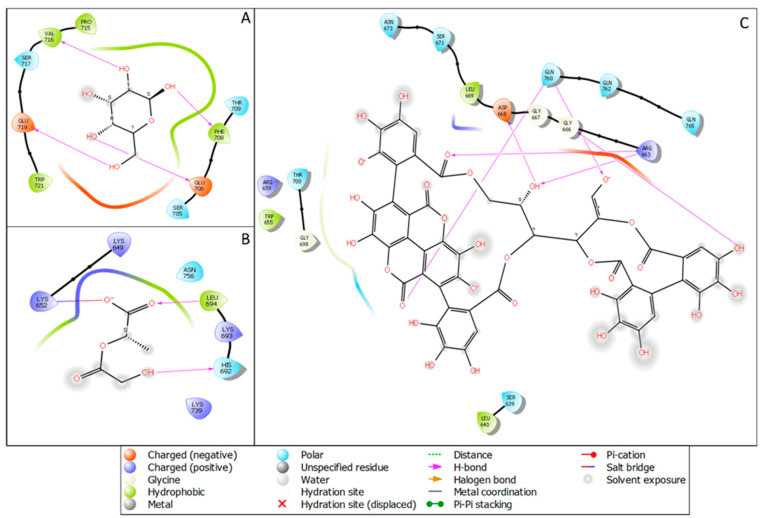
The ligand interaction diagram of the macrophage mannose receptor (1EGI) with (**A**) mannose, (**B**) PLGA and (**C**) punicalagin.

**Figure 3 molecules-27-06034-f003:**
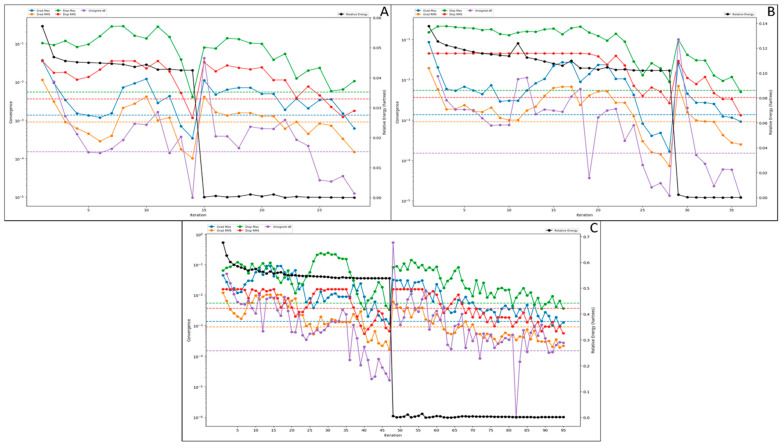
The quantum mechanics convergence for all three compounds: (**A**) mannose; 0 relative energy at 15 iterations, (**B**) PLGA; 0 relative energy at 30 iterations, and (**C**) punicalagin; 0 relative energy at 48 iterations.

**Figure 4 molecules-27-06034-f004:**
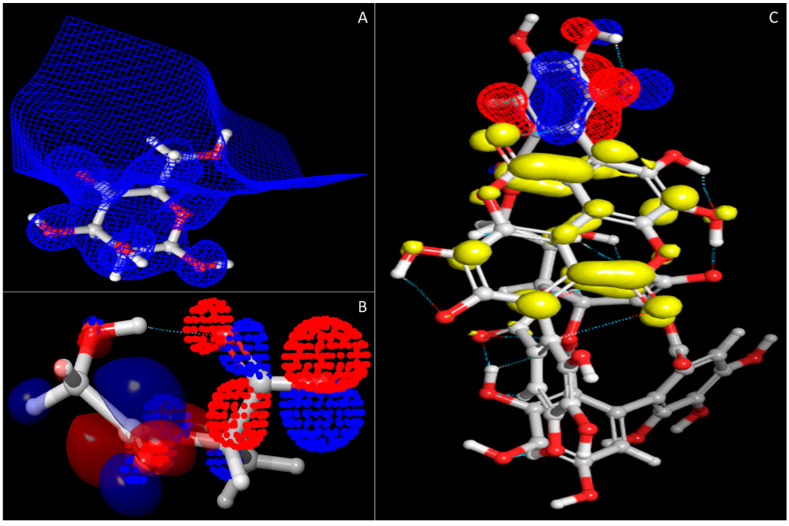
The HOMO and LUMO sites of all three compounds: (**A**) mannose; mesh is showing only the LUMO site as it does not have any HOMO site, (**B**) PLGA; dots are showing the HOMO site while the solid representation shows the HOMO site, (**C**)punicalagin; mesh shows the HOMO site while the solid yellow site shows the LUMO site.

**Figure 5 molecules-27-06034-f005:**
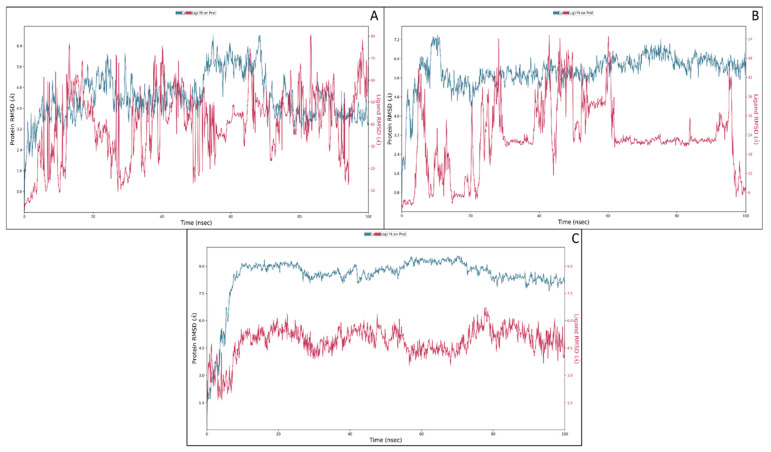
The root mean square deviation of the macrophage mannose receptor and (**A**) mannose, (**B**) PLGA, and (**C**) punicalagin.

**Figure 6 molecules-27-06034-f006:**
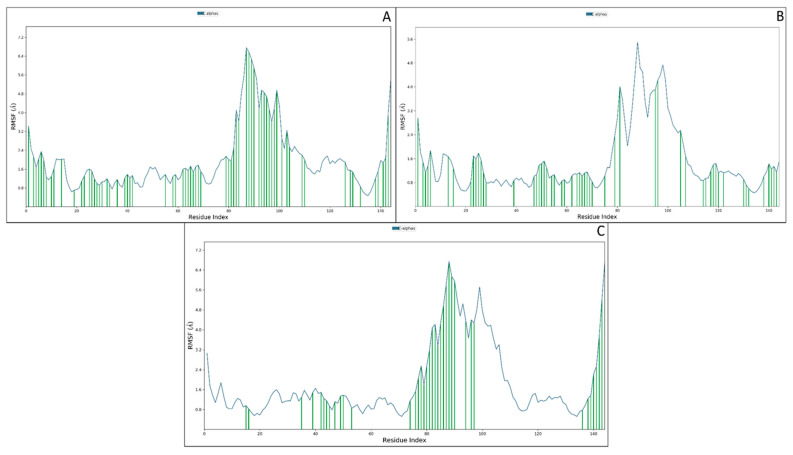
The root mean square fluctuations of the macrophage mannose receptor in the complex condition of (**A**) mannose, (**B**) PLGA, and (**C**) punicalagin. The blue line shows the amino acid level fluctuations, while the green vertical line shows the ligand contact with the amino acids at a particular simulation time.

**Figure 7 molecules-27-06034-f007:**
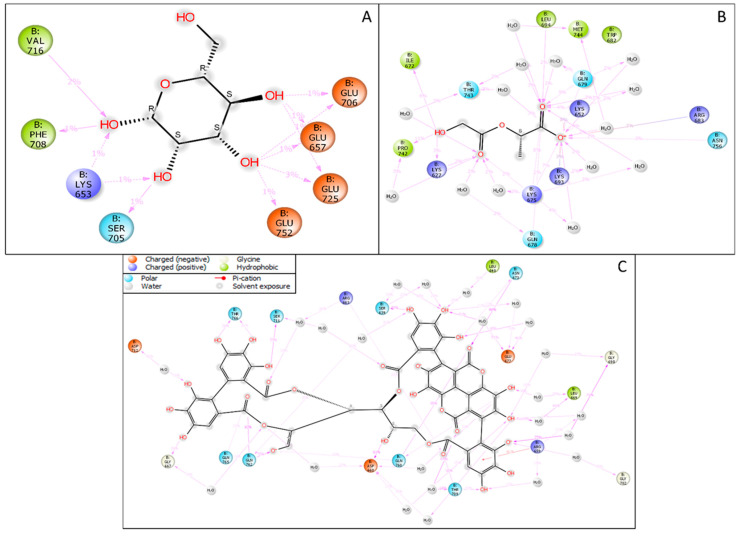
The interaction during the simulative periods among the macrophage mannose receptor in complex with (**A**) mannose, (**B**) PLGA, and (**C**) punicalagin.

**Figure 8 molecules-27-06034-f008:**
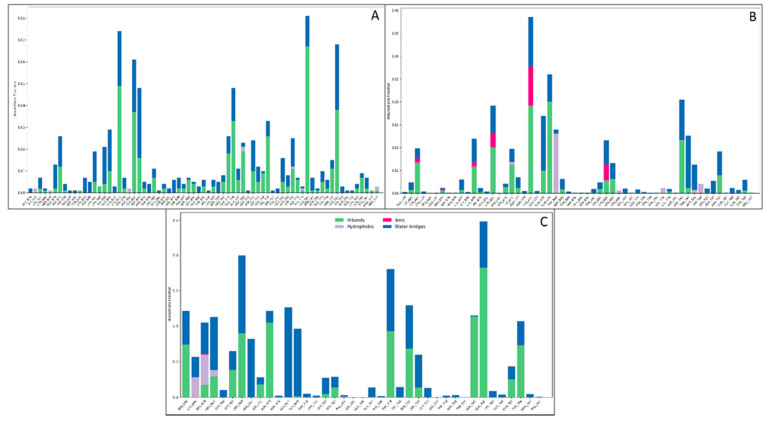
The histogram representation of the interaction count among the macrophage mannose receptor and (**A**) mannose, (**B**) PLGA, and (**C**) punicalagin.

**Table 1 molecules-27-06034-t001:** The docking score MM\GBSA score and other energies of the protein–ligand complex.

PDB ID	Ligand Name	PubChem CID	Docking Score	MM\GBSA dG Bind	Ligand Efficiency Sa	Ligand Efficiency ln	Glide Evdw	Glide Ecoul	Prime Hbond	H-Bond Count
1EGI	Mannose	18950	−5.811	−19.28	−1.109	−1.667	−6.965	−20.241	−68.76	4
1EGI	PLGA	23111554	−4.334	−18.38	−0.934	−1.312	−6.634	−17.529	−68	2
1EGI	Punicalagin	44584733	−2.931	−14.1	−0.161	−0.547	−31.984	−13.778	−70.84	6

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
