# Peer review of "Macrophage-Targeted Punicalagin Nanoengineering to Alleviate Methotrexate-Induced Neutropenia: A Molecular Docking, DFT, and MD Simulation Analysis"

_molecules, 2022, doi:10.3390/molecules27186034_

Round 1

Reviewer 1 Report (Previous Reviewer 1)

According to review’s comments, lots of revisions have been done. After reading the manuscript carefully, I think this manuscript could be published now.

Reviewer 2 Report (Previous Reviewer 2)

The paper is publishable in this revised form, although a better explanation of the model used for the nanocarrier is still needed. The complete Mn-PLGA-PG ligand interacting with the protein receptor is never simulated.

This manuscript is a resubmission of an earlier submission. The following is a list of the peer review reports and author responses from that submission.

Round 1

Reviewer 1 Report

This work was envisioned to predict the targeting efficiency of punicalagin (PG) nanoparticles to the macrophages. The research content is very important, but it is not suitable for publication at present. At least laboratory data validation is required before publication.

Reviewer 2 Report

Review Molecules-1832460

A nanomedicine-based approach that could propose a promising solution to the problems coupled with conventional drug delivery is needed. In this context, using Poly (lactic-co-glycolic acid) (PLGA) nanocarriers in drug delivery is promising. In the paper, a computational study is carried out to analyze some properties of the mannose decorated PLGA and punicalagin (Mn-PLGA-PG) nanoparticles. The significance of the content could be high, but there is still a lot of work to do for the paper to be publishable.

  1. The authors do not explain why chain A was not included in the in-silico receptor model.
  2. On page 2, the authors say that 'Therefore, in our proposed work, we explore the design of a targeted nanocarrier for bone marrow and whether the proposed nanocarrier is efficient in delivering the PG molecule to its site of action by molecular docking and dynamics simulation studies. Further, the study is in an experimental condition to design and validate the computational results.' Those two statements are rather pretentious because the calculations are reduced to model the ligand-receptor interaction. 
  3. Punicalagin is very different in size from the other two ligands. It is unclear to assert how this ligand fits the receptor's cavity. In addition, the complete nanocarrier is never docked to the receptor, which would be a better model. The authors dock the three different compounds separately without explaining why. 
  4. The molecular docking studies of mannose receptors with punicalagin, PLGA, and mannose have shown a silver lining to generate a nano-complex for all three ligands, and it can be provided as a dose for targeted therapy (page 4). The meaning of this sentence is unclear. 
  5. The energetic terms in Table 1 should be defined and analyzed. 
  6.  The QM optimizations of the different compounds are in the gas phase? The conclusions extracted from those calculations are useless for using those compounds in a nanocarrier. The utility of the HOMO-LUMO calculations is not shown either. 
  7. The study should be completed by docking and MD simulations of the self-assembled nanoparticle, otherwise the conclusions are not supported by the results.
  8. The title of section 3.2 should say Density Functional Theory.